# AudioUnlock: Device-to-Device Authentication via Acoustic Signatures and One-Class Classifiers

**DOI:** 10.3390/s25216510

**Published:** 2025-10-22

**Authors:** Alfred Anistoroaei, Patricia Iosif, Camelia Burlacu, Adriana Berdich, Bogdan Groza

**Affiliations:** Faculty of Automatics and Computers, Politehnica University of Timisoara, 300223 Timisoara, Romania; alfred.anistoroaei@aut.upt.ro (A.A.); patricia.iosif@student.upt.ro (P.I.); camelia.burlacu@student.upt.ro (C.B.); adriana.berdich@aut.upt.ro (A.B.)

**Keywords:** smartphone fingerprinting, microphone, loudspeaker, machine learning, acoustic signature

## Abstract

Acoustic fingerprints can be used for device-to-device authentication due to manufacturing-induced variations in microphones and speakers. However, previous works have focused mostly on recognizing single devices from a set of multiple devices, which may not be sufficiently realistic since in practice, a single device has to be recognized from a very large pool of devices that are not available for training machine learning classifiers. Therefore, in this work, we focus on one-class classification algorithms, namely one-class Support Vector Machine and the local outlier factor. As such, learning the fingerprint of a single device is sufficient to recognize the legitimate device and reject all other attempts to impersonate it. The proposed application can also rely on cloud-based deployment to free the smartphone from intensive computational tasks or data storage. For the experimental part, we rely both on smartphones and an automotive-grade Android headunit, exploring in-vehicle environments as the main area of application. We create a dataset consisting of more than 5000 measurements and achieve a recognition rate ranging from 50% to 100% for different devices under various environmental conditions such as distance, altitude, and component aging. These conditions also serve as our limitations, however, we propose different solutions for overcoming them, which are part of our threat model.

## 1. Introduction

The need to secure device-to-device (D2D) communication has significantly increased with the growth of the smart device market, which includes smartphones, smartwatches, ear pods, augmented reality headsets, and many others. This is also reflected by the ever-increasing number of research papers that address this topic. It is commonly known that microphones and loudspeakers have imperfections caused by the manufacturing process, making them unique. These differences come from how the transducer responds to sound pressure [1], as well as how the voice coil and the diaphragm manage to produce sound [2]. The chemical composition of the materials used in one production batch compared to the next, the wear of the manufacturing equipment, and variations in the electromagnetic properties of the driver give a uniqueness to each microphone and loudspeaker, as illustrated in [2]. These differences cannot always be discerned by the human ear, but signal processing techniques and machine learning algorithms can be used to detect such differences. Because of this, audio signals might be just the perfect way to deliver the right solution for D2D authentication.

The fact that most modern-day devices have microphones and loudspeakers already incorporated makes this choice a natural one. In the automotive context, this concept fits even better due to the fact that headunits have shifted in their main usage from basic radio functionalities to a whole package of services, like audio–video streaming from the internet, providing advanced features to users, remote control of the components, etc. Modern-day cars still use classical keys for authentication, while only a few high-end manufacturers are releasing solutions that use smartphones. But existing initiatives such as the Car Connectivity Consortium [3], where car and phone manufacturers are developing industry standards and solutions to use smartphones as car keys, are promising for the future of smartphones within the car ecosystem. So far, the main solution released by the aforementioned collaboration is based on NFC and UWB, but other options might be taken into account, which is a reason to conduct more research on alternative authentication sources. All of these come hand in hand with advances in the headunit hardware, which is able to perform more and more complex computations.

Another specific area of application may be improving multi-factor authentication (MFA), which requires the users to provide multiple pieces of evidence (factors) in order to prove their identity. Needless to say, users still heavily rely on passwords, and generally, when employed alone, these are far from being sufficiently secure (mostly because users are prone to choosing weak-entropy passwords). The premise of an MFA mechanism is the following: the more factors there are, the lower the probability that an attacker forges all tokens. These multiple factors might include something the user knows (like a password, a PIN), something the user possesses (like a code generated on a mobile device, a bank card), or something that the user physically possesses (a fingerprint, face shape, voice) [4,5]. In order to avoid security breaches, most big tech companies have already adopted MFA, e.g., Apple adopts MFA when checking users’ Apple IDs [6], Google and Facebook use MFA when users sign in with a new device [7,8], etc. Also, since the demand for such services is high and the implementation can be sensitive, there are even off-the-shelf solutions developed by Microsoft [9] and Google [10] for this purpose. Despite the fact that so far most of the industry’s solutions for MFA have been based on passwords or human fingerprints, the research community has investigated many other factors, like audio signals, for example. Whether using audio sounds to measure the proximity of devices, as in [11,12], or using ambient sound for authentication, as in [13,14], there are many other papers proving that audio signals could be a reliable and effortless addition to MFA.

The physical properties of the sound make it a good candidate for security: if broadcast at a low volume, which is enough for short distances, a malicious adversary that wants to intercept needs to be present near the devices or to use more intricate microphones. Indeed, there have been several recent works showing that sound can be intercepted in clever ways, even from a significant distance, but these kinds of attacks are beyond the scope of our work [15,16,17,18].

An overview of the system we designed is depicted in Figure 1. Within this system framework, we focus on the usage of one-class classifiers as the main solution. This seems to be generally neglected by related works which have used multi-class classification, and this creates a disparity with the real-world problem, where the task does not involve recognizing a single device from a larger set of multiple devices for which all fingerprints are available in the dataset but rather recognizing a single device from a larger pool of devices that are not available for training the classifiers. The target smartphone emits a synthetic sound (which has a short start frequency and afterwards a linear sweep signal) towards another smartphone or a headunit. The recording device processes this data, filters it, and interprets it (i.e., runs a machine learning model) either locally or on the cloud by sending over the normalized FFT frequencies for identification. Whether we discuss cloud or local data interpretation, we use a pickled machine learning model. This means that a Python 3.10 object hierarchy (in our case, a trained machine learning model) is converted into a byte stream file [19] (a pickled model or a .pkl file) which can later be loaded in different environments (in our case, an Android device or a cloud server). For identification purposes, we use the Fast Fourier Transform (FFT) to extract the frequency response from the linear sweep signal (generated by the loudspeaker and recorded by the microphone) and then use two one-class classifiers: Support Vector Machine (SVM) and the local outlier factor (LOF). The contributions of our work are the following:We employed the LOF classifier which, to the best of our knowledge, has not previously been used with microphone and loudspeaker data;We implemented an Android application that records sounds and relies on machine learning algorithms that run either on the smartphone or in the cloud, being able to differentiate between the smartphones that have emitted these sounds;We built a dataset of 1540 measurements using 22 different smartphones and one Android headunit in the process;We present concrete experimental results regarding the recognition accuracy and compare two classifiers against data from multiple devices, including same-model smartphones, using both a smartphone and a headunit to recognize the emitter;The experiments also address adversarial behavior, as well as environmental variations, like temperature or atmospheric pressure changes, which can degrade the classification accuracy.

**Figure 1 sensors-25-06510-f001:**
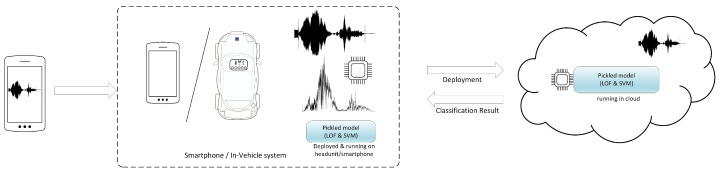
System overview.

The rest of this work is organized as follows. In Section 2, we present some of the related works. In Section 3, we present the implementation details alongside the devices we used. Section 4 provides the classifiers that we used.The experimental results are presented in Section 5, and the limitations are given in Section 6, while Section 7 contains the conclusion of our work.

## 2. Related Work

In the last few years, there have been quite a few works that have studied mobile device fingerprinting using different embedded sensors, e.g., from microphones and loudspeakers to cameras and accelerometers, etc. Here, we provide a brief overview of these works.

In a recent work, smartphones were fingerprinted by using the roll-off characteristics of the device’s loudspeakers [20]. For higher accuracy, they also relied on deep learning algorithms such as Convolutional Neural Networks (CNNs) and Bi Long Short-Term Memory (BiLSTM) networks. This work was closest to the current paper, but it differed in the selection of the classifiers, i.e., as already stated, the current work uses one-class classifiers, which represent a more realistic choice from a practical perspective. We also tested these classifiers on the public dataset made available by [20] to check the performance of the one-class classifiers but later relied on fresh experiments for the current paper. In contrast to [20], besides the distinct classifier, we also experimented with adversarial impersonation attempts, as well as with environmental variations, i.e., temperature and atmospheric pressure.

A method for generating a stable, unique, and stealthy device ID for smartphones, by exploiting the frequency response of loudspeakers, was described in [21]. For loudspeaker identification, the Euclidean distance between two samples was used, and in order to keep the process stealthy, frequencies between 14 kHz and 21 kHz were employed. The authors of [2] managed to fingerprint smartphones based on audio samples and considered three fingerprinting sources: loudspeakers, microphones, and joint speaker–microphone data. After recording audio data, Mel-Frequency Cepstral Coefficients (MFCCs) were used to localize spectral anomalies, while K-Bearest Neighbor (KNN) and Gaussian Mixture Models (GMMs) were the classifiers employed to identify the device.

There have been many other works that have used microphones for fingerprinting, the difference being the type of signals and the classifiers used. For example, the following research papers used synthetic sound as their signals but employed different classifiers, like Naive Bayes (NB) classifiers, SVM, and KNN in [22]; inter-class cross-correlation [23]; Maximum-Likelihood Classification [24]; and artificial neural networks [25]. Human speech was also used as the fingerprinting input for different classifiers like the MFCC [26], Linear Predictive Coefficients (LPCs), a Multi-Layer Perceptron classifier (MLP) [27], a Gaussian Mixture Model (GMM) [28], and Linear Frequency Cepstral Coefficients (LFCCs) [29]. Some of these works used preexisting datasets, while others managed to create their own.

A source smartphone identification system suitable for both clean and noisy environments was proposed in [30]. This system used the spectral distribution features of the constant Q transform domain and four classification techniques. The authentication of smartphones using the intrinsic physical properties of the mobile smartphones’ microphones using three classifiers, i.e., KNN, SVM, and a CNN, in the presence or absence of noises was studied in [31]. Three types of noises were considered: Gaussian white noise, Babble noise, and environmental noise from the street. The authors in [32] fingerprint 16 identical and 16 different smartphone microphones based on several in-vehicle noises, i.e., car horns, wiper sounds, hazard light sounds, and locomotive and barrier sounds. They also added several types of noise, i.e., traffic and market noises.

In [33], the authors used a few multi-class classifiers for source smartphone identification based on the encoding parameters of multiple audio files. Also, in [34], smartphone identification from recorded speech signals was elaborated with the goal of extracting intrinsic traces related to the smartphone used to record the data. Investigation of the self-localization problem in an ad hoc network of randomly distributed and independent devices in an open-space environment with low reverberation but heavy noise was carried out in [35]. The authors used acoustic emissions and computed the time of arrival for sounds to determine the distance between two smartphones, while landmark-based audio was used to synchronize audio recordings. In [36], different audio feature extraction algorithms, which aid in increasing the accuracy of identifying environmental sounds, were evaluated. Sound signals were detected using the standard deviation of normalized power sequences based on the zero crossing rate, MFCC, spectral flatness, and spectral centroid. Other papers have also proposed methods for defending against loudspeaker fingerprinting by enabling users to control the frequency response and modify the given stimulation signal [37].

Fingerprinting was also studied based on other smartphone sensors, such as cameras, accelerometers, gyroscopes, and others. For example, different works use distinctive features of the camera, like color filter array (CFA) [38], dark signal non-uniformity (DSNU) [39], and photo response non-uniformity (PRNU) [40]. Also, accelerometers were used to fingerprint devices in [24,41,42]. The possibility of fingerprinting devices based on combining sensors has been suggested in many papers, like [43], which used seven sensors (acceleration, magnetic field, orientation, gyroscope, rotation vector, gravity, linear acceleration), or [44,45], where two sensors were used (accelerometer and gyroscope). In other papers, software characteristics like the side-channel features from network traffic from several popular applications [46] or TCP [47] were used. Last but not least, the magnetic signals emitted by the CPU proved to also be able to fingerprint devices [48].

## 3. Implementation Details and Devices

In this section, we discuss the implementation details related to the Android applications, how we use the cloud’s computational capability, and a few statistical results regarding the recordings.

### 3.1. Microphone and Loudspeaker Details

Loudspeakers and microphones contain many components made of various materials. Therefore, due to imperfections in the manufacturing process, each loudspeaker and microphone will display slightly perceptible differences, just enough to make unique fingerprinting possible. Loudspeakers are electroacoustic transducers that convert electrical impulses into sound. Microphones carry out the opposite operation, converting sounds into electrical impulses. The vibration of a diaphragm is used to produce sound by a loudspeaker. The diaphragm is usually manufactured in the shape of a cone or dome, using different materials like paper, plastic, or metal. In loudspeakers, it turns the vibrations of the voice coil into sound waves, while for microphones, it turns the sound vibrations into movement of the voice coil.

### 3.2. The Android Application and Devices

To prove our concept, we developed two Android applications, one that handles sound recording and one that handles emission. Afterwards, we used the Python pickle module [19] to serialize the classifiers and extend the applications so that the models could run on smartphones. Further, we also managed to expand the functionality of the applications by connecting to a cloud server and running the classifiers on the cloud.

Figure 2 shows a flowchart that describes both the recorder and emitter applications. In the emitter, we specify a start frequency that the recorder later uses to detect whether transmission has started. Subsequently, the emitter starts sending the audio signal for 5 s. If the recorder accepts the authentication, the emitter stops; otherwise, the procedure is repeated. The recorder application is slightly more complex: in the user interface, the start frequency is specified (it can be selected by the user but must match the frequency of the emitter application). When the start frequency is detected, the raw values are added to an array for a duration of 5 s. Afterwards, the FFT is computed, and the values are normalized. These values are either sent to the cloud classification model or locally processed by the pickled model. A response is returned, indicating whether the smartphone was recognized or not. If the smartphone is recognized, the cloud classification model is regenerated to account for the newly collected data. Adding new data to the classification model is needed since the loudspeaker parameters may vary in time and the smartphone will become unrecognizable after longer periods (we will later show through experiments performed several months apart that this is indeed the case).

As an additional note, the sampling rate of the Android audio library is 44.1 kHz. Due to the Nyquist–Shannon sampling theorem, the maximum frequency that can be perfectly reproduced is 22.05 kHz. Alternatively, the sampling rate can also be set to 48 kHz, but we obtained better results for the former. In order to record signals, we used the Android AudioRecord class. For the Fast Fourier Transform (FFT), we used the JTransforms library [49]. The description of the FFT function, also implemented in this library, can be expressed in a mathematical way as follows: $X\left[k\right]=\sum_{n=0}^{N-1}x\left[n\right]\;e^{-j\;2\pi kn/N},k=0,1,\ldots ,N-1$. Here, x[n] are the time-domain samples, X[k] are the frequency-domain samples, *N* is the total number of samples, *j* is the index for the time domain, *k* is the index for the frequency domain, and *j* is the imaginary unit. We parsed the resulting FFT at intervals equal to 1 Hz and used only the highest amplitude in each interval. Thus, we obtained an array of 22,050 values.

In order to carry out more conclusive experiments, we used 22 smartphones from different brands and one Android headunit. We encountered problems with the headunit’s incorporated microphone, which was of poor quality. We tried to use an external microphone that could be acquired for the unit, but the results did not improve at all. Therefore, we decided to stick with the internal microphone for our experiments. In Table 1, we depict the list of devices, including their manufacturer and model, the label used in this work, and the number of devices of the same type. In order to increase the number of devices that we were using for measurements, we decided to add several iPhones as well. Since these smartphones do not run Android, we manually added the sound to be played on each device. For this, we used REW [50] to generate the desired frequencies, i.e., the linear sweep from 0 Hz to 20 kHz and the start frequency (this was set to 12 kHz for the headunit and 16 kHz for the smartphones). This linear sweep signal can be described by the following mathematical expression, $f\left(t\right)=f_{0}+\frac{\left(f_{1}-f_{0}\right)}{t_{1}}t$, where *t* denotes time, $f_{0}$ is the start frequency at time 0, and $f_{1}$ is the instantaneous frequency at time $t_{1}$. Concretely, in our implementation, we used $f_{0}=0$ Hz, $f_{1}=20$ kHz, and $t_{1}=10$ s. This led to a wav file which could be played on iPhones as well in order to check the identification rate from the recorder.

In order to have a more scalable system, we decided to extend it with cloud processing features. Concretely, we ran the classification model on a cloud server provided by Heroku [51], while the smartphone (the recording application) was used only to collect data and display the result. The receiver application, after extracting data with the algorithm described before, connects to the cloud server using the Volley HTTP library [52] developed by Google. Data is encapsulated in a JSON object sent through a POST request. On the cloud side, Heroku offers the possibility to be configured through a Github deployment pipeline. Using Flask [53], we managed to run our pickled model in the cloud and return a response to the client (i.e., the receiver application on the smartphone), indicating whether the data belonged to the target smartphone or not.

### 3.3. Preliminary Statistical Results Regarding the Recordings

In order to see whether there are significant statistical differences between the devices, we choose to extract several spectral statistical characteristics. For each device in our experiments, we perform 50 measurements and extract the mean value, which we present as a box plot in Figure 3. The characteristics we have taken into consideration are
The power of the signal in a selected frequency band (powerband) refers to the average power of the input audio signal in the 20 Hz–20 kHz band;The standard deviation of the spectrum (Std)—how much the frequencies deviate with respect to the mean value;The peak of the signal (Pks)—the maximum values of the input audio signal;The average value of the signal (Mean)—more precisely, the average value of the power spectrum of the audio signal;The spikeness or flatness of the distribution compared to the normal distribution (kurtosis) represents a statistical parameter that quantifies the distribution shape of the audio signal compared to the Gaussian distribution, i.e., showing whether the distribution is sharply peaked or not;The ratio of skewness to the standard deviation (skewness), which represents the asymmetrical spread of the audio signal about its mean value.

**Figure 3 sensors-25-06510-f003:**
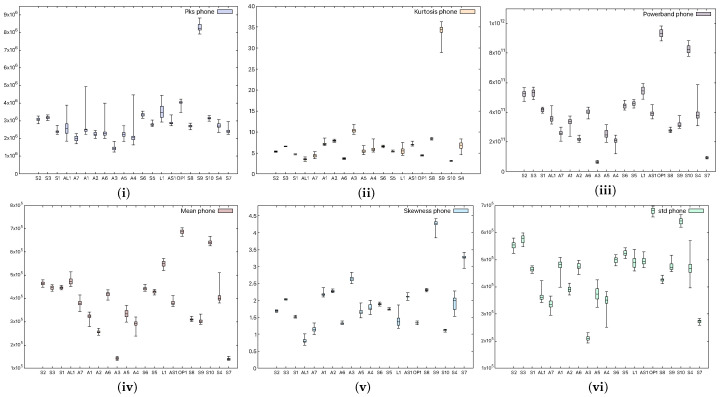
Statistical characteristics, (**i**) signal peak, (**ii**) kurtosis, (**iii**) powerband, (**iv**) mean, (**v**) skewness, and (**vi**) standard deviation, in smartphone (device M1) records.

Figure 3 shows the aforementioned statistical coefficients when a smartphone was used as the receiver (device M1), while Figure 4 shows the same characteristics when a headunit was used as the receiver (device E1). From the figures, it is clear that the devices can be easily distinguished, and moreover, the recordings performed with the headunit show larger boxes in the plot, which are due to a less accurate measurement. Indeed, as the selected classifier later suggests as well, the confusion rate between devices is higher with the headunit because of the poor performance of the microphone.

## 4. Classifiers and Noise Resilience

In this section, we detail the classifiers that we used, along with some preliminary results on an existing dataset. We also consider the impact of noise on these classifiers, as noise will be an important factor in practical scenarios.

### 4.1. Selected Classifiers

The problem we are trying to tackle is the use of one-class classification, which allows us to learn one specific device and later test the model against the other devices. As stated, most existing works use multi-class classification techniques, where all devices from the dataset are used in both the training and testing phases. This approach is not sufficiently realistic, and to improve this, we decided to use one-class classification techniques, namely two such classifiers, as described next:

#### 4.1.1. One-Class SVM (OC-SVM)

OC-SVM [54] is an unsupervised algorithm for learning a decision function for novelty detection first introduced in [55]. SVMs do not represent a probability distribution. Instead, their approach is finding a positive function to model regions with a high density of points and a negative function for low-density regions. All data points are separated from the origin in the feature space using hyperplanes, and the points are projected onto a higher-dimensional space. In contrast to typical SVMs, it employs a parameter to set the outlier proportion in the data, maximizing the distance between the hyperplane and the origin. Outliers are points that are below the hyperplane and closer to the origin. The absence of target labels through the training phase for OC-SVM is another distinction.

OC-SVM requires the following parameters to be specified: tol (the tolerance needed to stop), kernel (the type of kernel that will be used by the algorithm), degree (the degree of the polynomial function), and nu (higher and lower bounds on the percentage of margin errors and, respectively, the percentage of support vectors in relation to the total number of training samples). We chose kernel = ‘rbf’, degree = 5, tol = 1 $\times \;10^{-7}$, and nu = 0.001. A kernel is required because it allows for the projection of samples from the original space to the characteristic space. The Radial Basis Function (RBF) is one of the most used kernels in the literature. It aids the algorithm in determining the radius of the hypersphere, as well as the polynomial functions. We used a large degree to make the algorithm more precise. The tolerance parameter, tol, and the nu parameter are set to such low values to reduce the possibility of running into many incorrect predictions.

#### 4.1.2. Local Outlier Factor (LOF)

The local outlier factor (LOF) [56] is another unsupervised algorithm, designed for outlier detection, which was first introduced in [57]. It also has density at its core and aims to identify local outliers, i.e., points that are not fundamentally different from the rest of the population but deviate significantly from their neighbors. As far as density is concerned, this means that a local outlier is an example that greatly differs in density from the surrounding points. The algorithm can be configured using the following parameters: the number of neighbors considered for each sample, the distance metric used, and the contamination ratio of the data, as well as a boolean value that controls whether the algorithm is being used for novelty detection.

For each point, the algorithm computes a score (the local outlier factor), i.e., a degree that quantifies how much a point resembles an outlier [57]. Generally, a sample is considered an outlier if its corresponding LOF is above a certain threshold. The score itself is determined by looking at the local reachability distances (LRDs) of the point of interest versus that of its K-nearest neighbors [57]. To define the LRD, a brief understanding of K-neighborhoods is needed. A K-neighborhood contains the points that lie within a circular region whose radius is equal to the distance between the point of interest and its K-th nearest neighbor (i.e., the K-distance). Similarly, the reachability distance can be defined either as the K-distance for points that lie within the K-neighborhood or the actual distance between the point that defines the neighborhood and a target point for samples outside of this region. The LRD of a point is then computed by averaging the reachability distance over the points in its K-neighbor and subsequently taking the inverse of this average.

As was the case for One-Class SVM, we use the scikit-learn implementation [58,59], choosing the following parameters: n_neighbors = 19, metric = ‘manhattan’, contamination = 0.3, and novelty = True. Since we are using the algorithm for novelty detection, training is carried out entirely on samples belonging to one known class. The literature implies that in this case, the training data should be deemed completely uncontaminated [60]; however, our experiments suggest that setting a low level of contamination, i.e., contamination = 0.3, improves the overall performance on our datasets. This could be explained by the fact that during data collection, a certain level of environmental noise may be introduced into the samples. As for the metric used, i.e., the Manhattan distance, this was preferred for our high-dimensional data since it sums over the absolute value of the difference between each scalar, i.e., $\sum_{i=1}^{n}\left|x_{i}-y_{i}\right|$. Other works have also recommended it for high-dimensional data [61]. The number of neighbors was empirically determined.

### 4.2. Testing Classifier Resilience Against Background Noise

In order to test the reliability of the classifiers in the context of device fingerprinting, we decided to use a public dataset which contained audio signals from 16 Samsung Galaxy J5 loudspeakers with and without noise [20]. Concretely, the type of noise is additive white Gaussian noise (AWGN) overlapped using Matlab R2020b.

The first use case tested was audio signals without noise. For this, we trained the classifiers with half of the available data for each device (50 samples) and tested them on the other half. The results are presented in Figure 5, and as can be seen, the identification success rate is between 54 and 100%. Afterwards, we focused on noise-affected data. We trained the classifiers with half of the noise-affected samples and tested them against the rest. The results are also presented in Figure 5. As can be seen, the identification success rate is between 50 and 100%.

In both cases (with and without noise), the SVM classifier provides better results than those of the LOF classifier. For example, SVM has an average recognition rate for both cases of around 89%, while for the LOF, this value is 70%. Comparing the classification results with and without noise, we observe that the influence of noise is the same on both classifiers. For example, the performance of the LOF classifier decreases from an average value of 72% (without noise) to an average value of 67% (on data with noise). This means that 5% is lost for the identification rate when noise is added. For SVM, the performance decreases from an average of 93% to an average of 86% when adding noise to the data, which equates to a 7% decrease in the identification rate. To sum up, both one-class classifiers perform well, but SVM is better on existing datasets.

## 5. Experimental Results

In this section, we present the experimental methodology along with various details, such as the placement of the devices and the influence that distance has on measurements.

### 5.1. Off-Line Measurements and Results

We tested the classifiers based on pre-recorded data stored in the frequency domain as FFT coefficients. We performed 50 measurements with each smartphone, having another smartphone as the recorder (in this case, M1). A measurement consists of detecting the start frequency and then the linear sweep that follows for 5 s; it contains all frequencies in the range of 0 Hz to 20 kHz. During the experiments, the smartphones and headunit were, respectively, placed at a distance of 10 cm, with their loudspeakers facing the microphone of the recorder. This is also shown in Figure 6, which depicts one of our smartphones during the recording process.

Of the 50 measurements, 25 were used to train the classifiers and generate the pickled model. Then, we evaluated each model on the other 25 measurements. The results are presented in Figure 7. As a side note, in the corresponding table, N/A marks the measurements that were impossible to perform because the recording and the emitting device were the same (i.e., device M1). The lowest recognition rate is 52%, while the highest is 92%. These results are influenced by many factors, but the microphone quality is the most critical. This can be observed easily in Figure 7 since the measurements made with the headunit are slightly worse than those performed with the smartphone. Indeed, the average recognition rate for the smartphone recordings is 75.80% (SVM and LOF), while for measurements performed with the headunit, the average percentage decreases to 69.36%. By comparing the classifiers, we observe that in cases when the recordings were performed with the smartphone, the SVM has slightly better results than those of the LOF, having a 77.90% average recognition rate compared to 73.71%. The difference between these two classifiers decreases when the recordings are made with the headunit due to the poorer quality of the microphone, i.e., 70.36% for LOF and 68.36%, respectively, for SVM.

We note that the signals emitted by the smartphones were quite distinct, even for the same type of devices. For example, as can be seen in Figure 8, for the same smartphone model (devices S5, S6, and S7, which are all copies of the same Samsung Galaxy J5), the emitted sound was quite different in both the time and frequency domains. For different models (S9 and A4), the differences are even more obvious. Such differences also prevent impersonation attacks, as we discuss later in the paper.

### 5.2. Live Measurements and Results

We now test the classifiers based on live attempts to authenticate the smartphone to the headunit. For this, we run the classifiers directly on the smartphones or in the cloud, using the pickled models. All operations mentioned above, especially on the recorder app side, were not time-consuming. Concretely, the computational costs introduced by each step are as follows: (i) write the output to a .wav file (required for off-line analysis only): 0.12 s; (ii) apply the FFT: 0.39 s; (iii) write the FFT-normalized values to a file: 0.58 s; (iv) response from the model running on the smartphone: 0.01 s; (v) response from the model running on the cloud server: 1.23 s. These measurements were computed as the average values after 10 attempts.

To sum up, after finishing emitting the linear sweep (5 s), the output (if the smartphone is recognized or not) was provided in about 1.6 s if the model was run locally (on a smartphone) or about 2.5 s if the cloud server was involved. The delays were higher for the cloud-based solution because of network timings. However, the cloud solution may facilitate more complex algorithms and a centralized solution for authenticating various devices. The cloud may also make our solution more scalable by supporting older devices or Internet of Things (IoT) with low processing power. Also, updating or retraining the machine learning models would be easier and more efficient in the cloud.

To check whether the authentication results of the classifier executed on the smartphone/cloud were similar to those obtained locally in the previous section, we performed another set of 10 measurements involving all of the smartphones. The results are presented in Table 2. By comparing the classifiers, we show in Figure 7 that the SVM and LOF classifiers have almost the same recognition rate, concretely, 78.57% for the LOF and 78.09% for SVM (note that in this case, fewer smartphone-based experiments were conducted). With the headunit, the LOF classifier has a 71% average recognition rate, while SVM drops to 64%. This is in contrast with the results obtained when using off-line measurements, where the SVM classifier showed better results than those of the LOF classifier.

## 6. Limitations and Discussion

In this section, we introduce the threat model and discuss the main limitations of our system.

### 6.1. Threat Model: Device Impersonation by Replaying Sounds

The main adversaries that we consider are external devices that are able to record and attempt to reproduce the sound emitted by the legitimate device. Successfully penetrating the system would grant the adversary complete access to the services provided to the targeted user. We consider that devices that are too far from the system are less likely to produce good recordings, so the distance that we take into account for the adversary is below 2 m. Consequently, we also tested the application against a targeted impersonation attack. For this, we recorded the frequencies emitted by several devices and we played them from other devices to check whether any device impersonation occurred. Such impersonation attempts consistently failed due to significant differences between the recorded sound and the original signal. For example, as shown in Figure 9, when we record the sound emitted by device S1 with device S7 and later play the recording, we can see that the amplitudes are significantly higher than those of the original signal. Also, when we record the sound emitted by device S1 with device S9 and later play the recording, we can see that the signal pattern is completely different from the original one, despite the amplitudes being almost the same. Since the overall shape of the signal is different, it is very easy for the classification algorithms to repel the impersonation attack.

### 6.2. Distance’s Influence on the Recognition Rate

It is obvious that the distance between devices has an impact on the identification accuracy. The authors of [62] also provide rigorous equations for the acoustic attenuation, exhibiting a power-law, better described by $S\left(x+\Delta \left(x\right)\right)=S\left(x\right)e^{-\alpha (\omega )\Delta (x)},\alpha \left(\omega \right)=\alpha_{0}\omega^{\eta }$. Here, S is the amplitude of an acoustic field variable such as velocity or pressure, Δ(x) is the wave propagation distance, α(ω) is the attenuation coefficient, and ω is the angular frequency, while η and $\alpha_{0}$ are material-dependent parameters, empirically obtained.

To test the impact of distance, we used four devices: S6, AS1, A2, A4. The first thing we did was collect 10 samples for certain fixed distances between devices, i.e., 20 cm, 40 cm, 80 cm and 160 cm. In addition, we tested two scenarios. Firstly, both classifiers (the LOF and SVM) were trained with half the samples and tested on the other half. In this case, the recognition rate did not decrease with the distance, but rather, it remained mainly steady for all distances. The results are similar to the case referenced in Figure 7(i) and comparable with those in Figure 7(ii). This can be explained by the fact that if we retrain the classifier with new audio signals affected by the distance, it will recognize signals originating from the same distance. Secondly, we used the classifier already trained with data from Section 5.1, and we observed that the recognition rate decreased from over 70% for both classifiers with devices placed at 20 cm to 0% when the devices were moved 160 cm apart, as can be seen in the bar charts from Figure 10(i). This is expected since the audio signal is affected by the distance, and thus, the device is harder to recognize. In practice, we consider this distance more than enough to pair two devices (for example, the distance inside a car between the microphone and any passenger’s smartphone will be less than 160 cm, if we take into account the automotive scenario). In Figure 10(ii,iii), we also present plots in both the time and frequency domains to show the impact of distance.

### 6.3. Environmental Influence on the Recognition Rate

Device aging affects both microphones and loudspeakers, as well as any other component, thus resulting in slight changes in the audio waveform produced/captured by the smartphone. Since microphones and loudspeakers are electromechanical devices, that is, they have moving components whose elasticity changes with time (e.g., foam or diaphragm), they are even more affected than simple electronic components. We observed this empirically too by using the machine learning models with data obtained 7 months apart. In this case, the smartphones were not recognized any more; that is, the data collected 7 months after the original training data no longer matched. Figure 11(i) shows the differences in the recorded data for the A4 device after 7 months. By reducing this interval to 1 month after training, all devices were recognized. This justifies our choice, already discussed in the flowchart of our application, to add each new sample that was correctly identified to the training dataset. After longer periods of inactivity, in practical scenarios, a user will likely have to rely on other authentication factors in order to update the fingerprint of the smartphone or to reauthenticate the device. In fact, this recommendation is already included in some security standards or draft recommendations from the industry, such as NIST SP 800-63B [63], OWASP [64], ISO/IEC 27001 [65], ISO/IEC 27002 [66], etc. In this context, our cloud component proposal can help by regenerating the model each time a smartphone is recognized. The training dataset will grow with time, but the cloud can handle it and will contain the latest signals.

Besides aging, there are also other factors that influence the device’s components. For this, we verified whether lower temperatures could also affect the results. Therefore, the smartphones were placed in a refrigeration environment for about 30 min, and we tested the behavior at various temperatures. At 8 °C, 14 °C, and 18 °C, the devices were easily recognized. However, when we lowered the temperature to 2 °C, the models were not able to recognize the smartphones any more. At this temperature, it can be seen in Figure 11(ii) that the waveform is quite different to what we recorded at room temperature. This suggests that if high temperature variations are to be expected, then dedicated models have to be trained. Some smartphones, depending on how the Linux kernel is configured, provide access to the thermal zone temperature files [67] which contain the temperature of the smartphone’s CPU. This can be used to enable specific models according to the CPU temperature, but this goes beyond the scope of our work.

Another factor that might impact the recognition rate is atmospheric pressure. The first measurements were made by us at about 300 m above sea level, and we managed to perform another round of measurements at about 1300 m above sea level. The recognition rate was the same, so the variation of 1000 m in altitude, which corresponded to 1.56 psi, had no impact. The signal remained almost identical in both cases, as can be seen in Figure 11(iii). According to [68], altitudes above 3600 m have a noticeable effect on sounds, caused by atmospheric pressure. We were unable to conduct experiments in such conditions since we did not have a hyperbaric chamber at our disposal.

The calibration of the models for variations caused by aging, temperature, or pressure requires a larger amount of data and a longer processing time; thus, the cloud extension of the proposed system facilitates rapid regeneration and testing in the cloud. We evaluated the computational time and obtained an average of 13.5 s when using Google Colaboratory [69] for training on approximately 550 samples. The first run, during which the necessary packages were loaded and the drive was mounted, took 71.5 s.

## 7. Conclusions

Our results indicate a good recognition rate for smartphones, with an average of 77.90% for the LOF algorithm and 73.71% for SVM when running on smartphones, but this may be influenced by multiple factors. One such factor is the quality of the microphones: while for most smartphones the microphones were of good quality, the Android headunit had a decreased recognition rate due to the poor quality of its microphone. This issue can be mitigated by using an external microphone of better quality. Another factor that impacts the results is the distance between microphones and loudspeakers. We performed most experiments with the devices placed 10 cm apart and increased the distance up to 160 cm, until the microphone was no longer able to record the start frequency. Given the nature of the scenario, it is easy for users to place the devices close to one another, so distance should not be a concern. Clearly, a bigger concern is the aging of the device and the effect of temperature or even altitude that we discussed in the last section. In this respect, the most significant concern, shown through our results, is aging, as measurements taken several months apart may lead to serious classification errors. Continuously updating the device fingerprint seems to be the only means to circumvent this issue.

Regarding the two algorithms that we used (SVM and the LOF), we noticed that when using a good microphone, SVM had slightly better results, while with the poorer-quality microphone, the LOF gave somewhat better results. The two algorithms that we used had no problems in distinguishing between identical devices—in our experiments, we used three identical Samsung J5s, and the algorithms did not have any problems distinguishing between them. We also tried to impersonate a device by recording its output with another device and replaying the recording, but the system was resilient against such attacks.

Our investigations gave promising results, but clearly, the practical adoption of such a system, especially in the automotive context, requires more investigations, which we leave as potential future work.

## Figures and Tables

**Figure 2 sensors-25-06510-f002:**
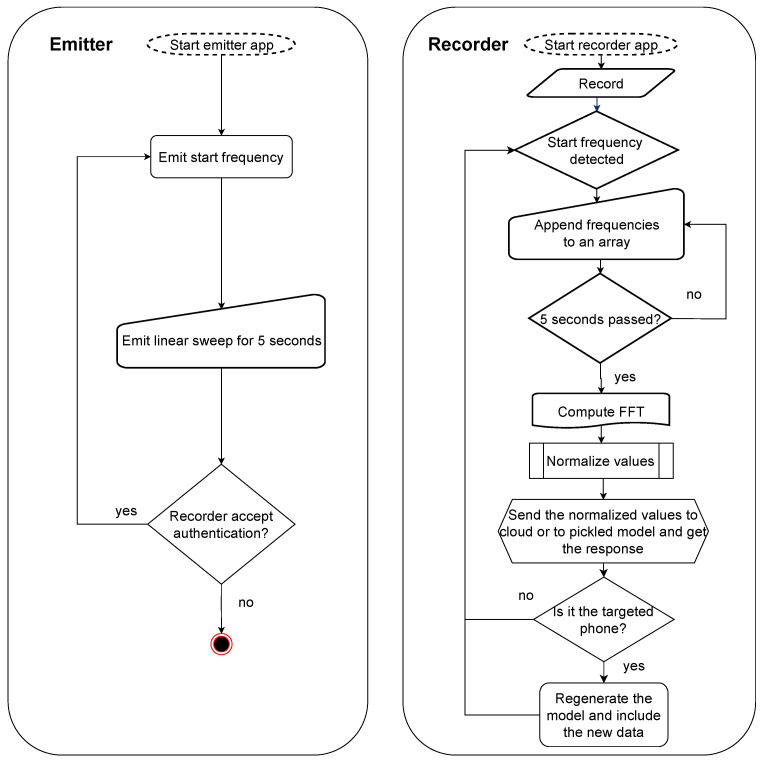
Flowchart of the emitter and receiver apps.

**Figure 4 sensors-25-06510-f004:**
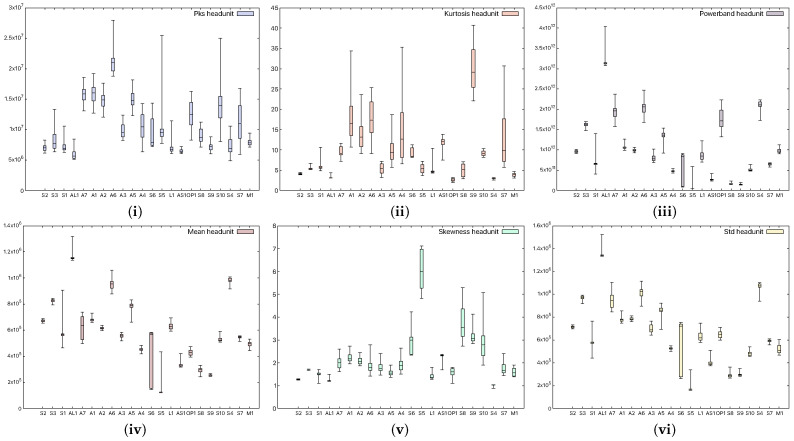
Characteristics extracted, (**i**) signal peak, (**ii**) kurtosis, (**iii**) powerband, (**iv**) mean, (**v**) skewness, and (**vi**) standard deviation, from headunit records.

**Figure 5 sensors-25-06510-f005:**
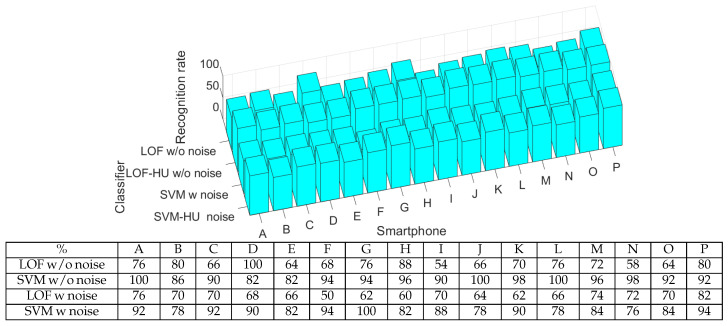
The acceptance rate for each smartphone as bar charts (**up**) and numerical values (**down**) using the existing dataset.

**Figure 6 sensors-25-06510-f006:**
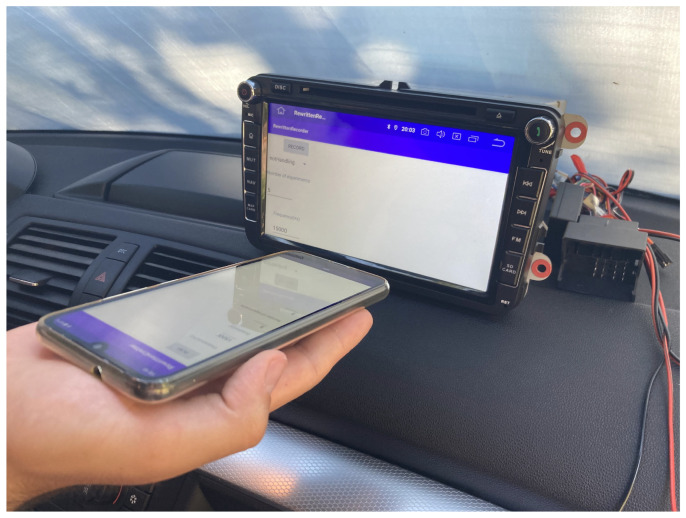
Smartphone and headunit placement.

**Figure 7 sensors-25-06510-f007:**
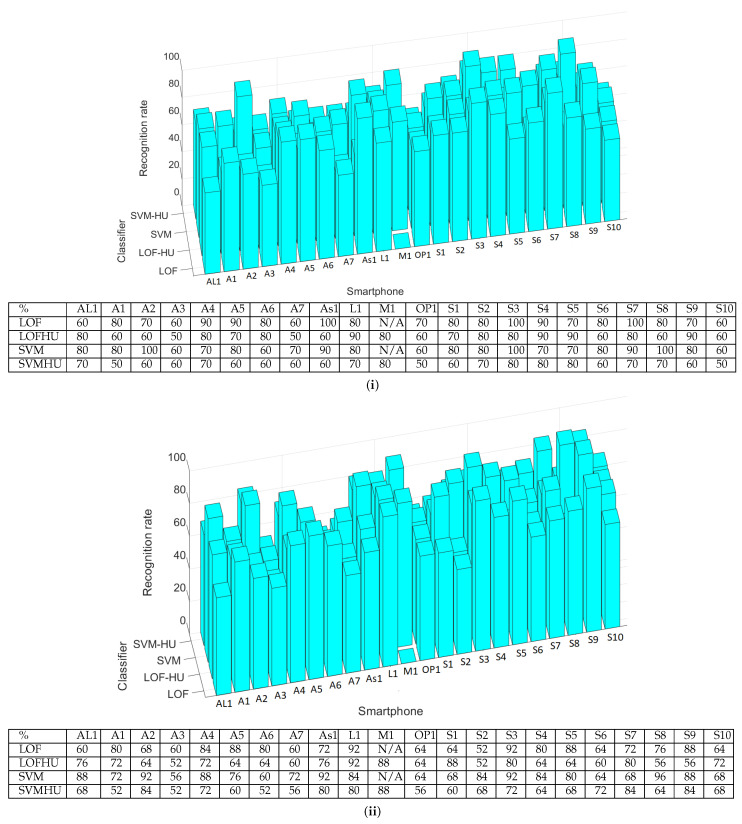
Identification percentages for each smartphone as bar charts and numerical values. (**i**) Results for 10 measurements with classification performed in the cloud; (**ii**) results for 50 measurements, 25 used for training and 25 for testing, with classification performed on the local device.

**Figure 8 sensors-25-06510-f008:**
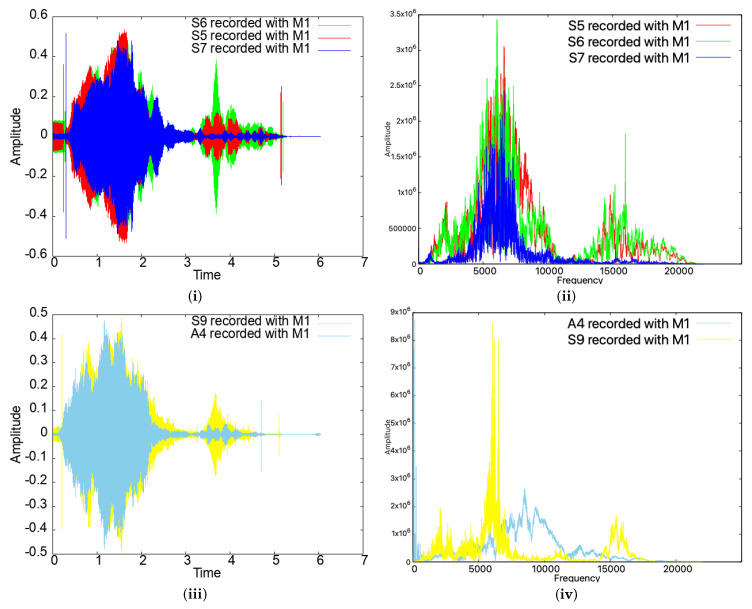
Time- and frequency-domain representations of the sweep signal for three identical devices (**i**,**ii**) and for two different devices (**iii**,**iv**).

**Figure 9 sensors-25-06510-f009:**
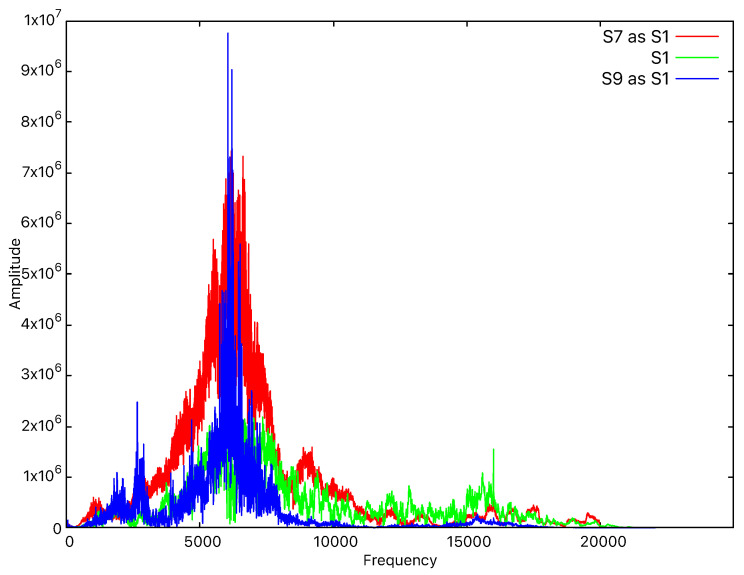
Frequency-domain representation in the case of S1 impersonation.

**Figure 10 sensors-25-06510-f010:**
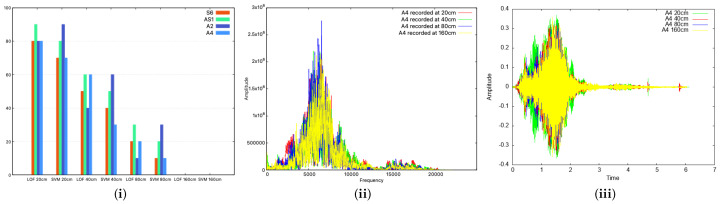
Identification rate along the distance as bar charts (**i**): distance’s influence in the frequency domain for the A4 smartphone (**ii**); distance’s influence in the time domain for the A4 smartphone (**iii**).

**Figure 11 sensors-25-06510-f011:**
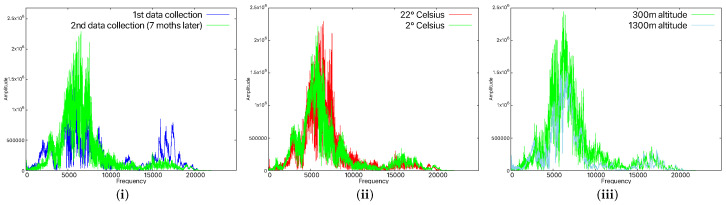
Time-domain representation for 7-month time difference for the A4 smartphone (**i**), time-domain representation in the case of different temperatures for the A4 smartphone (**ii**), and time-domain representation in the case of different altitudes for the A4 smartphone (**iii**).

**Table 1 sensors-25-06510-t001:** Devices from our experiments.

Brand	Model	Label	No. of Devices
Allview	Viper V1	AL1	1
Apple	iPhone 5s	A1 & A2	2
	iPhone SE1	A3	1
	iPhone SE2	A4 & A5	2
	iPhone 8	A6	1
	iPhone 13 mini	A7	1
Asus	Nexus	AS1	1
Erisin	PX5	E1	1
LG	Stylus 2	L1	1
Motorola	E6Plus	M1	1
OnePlus	7T	OP1	1
Samsung	Galaxy A3	S1	1
	Galaxy A10	S2 & S3	2
	G386F Galaxy Core	S4	1
	Galaxy J5	S5 & S6 & S7	3
	Galaxy S6	S8	1
	Galaxy S7 Edge	S9	1
	Galaxy Tab S7	S10	1
Total			23

**Table 2 sensors-25-06510-t002:** Success rate for 10 measurements with classifiers running on smartphones/headunit.

Nr.	Device	LOF Phone	LOF Headunit	SVM Phone	SVM Headunit
1.	AL1	60%	80%	80%	70%
2.	A1	80%	60%	80%	50%
3.	A2	70%	60%	100%	60%
4.	A3	60%	50%	60%	60%
5.	A4	90%	80%	70%	70%
6.	A5	90%	70%	80%	60%
7.	A6	80%	80%	60%	60%
8.	A7	60%	50%	70%	60%
9.	AS1	100%	60%	90%	60%
10.	L1	80%	90%	80%	70%
11.	M1	-	80%	-	80%
12.	OP1	70%	60%	60%	50%
13.	S1	80%	70%	80%	60%
14.	S2	80%	80%	80%	70%
15.	S3	100%	80%	100%	80%
16.	S4	90%	90%	70%	80%
17.	S5	80%	60%	80%	60%
18.	S6	100%	80%	90%	70%
19.	S7	70%	90%	70%	80%
20.	S8	80%	60%	100%	70%
21.	S9	70%	90%	80%	60%
22.	S10	60%	60%	60%	50%

## Data Availability

The raw data supporting the conclusions of this article will be made available by the authors on request.

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
