# Peer review of "AudioUnlock: Device-to-Device Authentication via Acoustic Signatures and One-Class Classifiers"

_sensors, 2025, doi:10.3390/s25216510_

Round 1
Reviewer 1 Report
Comments and Suggestions for Authors
This paper proposes an inter-device authentication method based on acoustic features and one-class classifiers, which is innovative and practical, but there are also some shortcomings:
1) The novelty and main contribution of this paper are not obvious, the author should explain why this research is carried out.
2) The threat model is lack of description in this paper.
3) This paper lacks mathematical or formal justification to demonstrate the effectiveness of the proposed scheme.
4) In the experimental part, due to the aging of equipment, the change of acoustic features leads to a significant decline in recognition rate. Although the dynamic update strategy is proposed, the redundancy problem of long-term dependence on other authentication methods is not solved.
5) As shown in Figure 12, acoustic signal is distorted and identification fails completely at low temperature, but the author does not propose temperature compensation or multi-model switching scheme. In addition, as shown in Figure 10, the recognition rate drops to 0% when more than 160cm is exceeded, which causes the scheme to be less feasible in practice.
6) Lack of further proof of security, how will it cope when malicious devices tamper with acoustic signatures through physical contact? For example, are adversarial attacks that currently only test simple recording playback resistant to generating adversarial examples or signal tampering?
Comments on the Quality of English Language
The English could be improved to more clearly express the research.
Author Response
Reviewer#1, Concern #1: 1) The novelty and main contribution of this paper are not obvious, the author should explain why this research is carried out.
Author response: We thank the reviewer for expressing this concern. We have added the following text in order to clarify that the main motivation behind the paper is using one-class classifiers, which was not done in previous works, though this is much more realistic given the application setup:
Within this system framework, we focus around the use of a one-class classifier as the main solution. This seems to be generally neglected by related works which use multi-class classification, but this creates a sort of disparity from the real world problem in which the task is not to recognize a single device from a larger set of multiple devices, having all these fingerprints in the dataset, but to recognize a single device from a larger pool of devices that are not available for training the classifiers.
Besides this, several contributions were already outlined on page 3, line 92:
The contributions of our work are the following:
- we employed the LOF classifier which, to the best of our knowledge, was not previously used with microphone and loudspeaker data,
- we implemented an Android application that records sounds and relies on machine learning algorithms that run on the smartphone or in cloud, being able to differentiate between the smartphones that emitted these sounds,
- we built a dataset of 1,540 measurements using 22 different smartphones and one Android headunit in the process,
- we present concrete experimental results regarding recognition accuracy and compare
two classifiers against data from multiple devices, including same model smartphones, using both a smartphone and a headunit to recognize the emitter,
- the experiments also address adversarial behavior as well as environmental variations, like temperature or atmospheric pressure changes, which can degrade the accuracy of the classification.
Reviewer#1, Concern #2: 2) The threat model is lack of description in this paper.
Author response: We thank the reviewer for this observation. In section 6.1, we now clarify the adversary model that was considered in our work. The following text was added to the paper:
The main adversaries that we consider are external devices that are able to record and attempt to reproduce the sound emitted by the legitimate device. Successfully penetrating the system would grant the adversary complete access to the services provided to the targeted user. We consider that devices that are too far from the system are less likely to make good recordings so the distance that we take into account for the adversary is below 2 meters.
Reviewer#1, Concern #3: 3) This paper lacks mathematical or formal justification to demonstrate the effectiveness of the proposed scheme.
Author response: We thank the reviewer for raising this important point. The formalism regarding the function that generates the sweep is now outlined on page 6, the Fast Fourier Transform on page 5, and the linear attenuation that was already presented on page 15. We note that the effectiveness of such approaches, based on physical characteristics, are usually assessed based on experimental measurements and we were unable to find other mathematical justifications in related works.
The following text was added to the paper:
The description of the FFT function, also implemented in this library, can be expressed in a mathematical way as follows: $
X[k] = \sum_{n=0}^{N-1} x[n] \, e^{-j \, 2\pi kn / N}, \quad k = 0, 1, \dots, N-1 $ where x[n] are the time-domain samples, X[k] are the frequency domain samples, N is the total number of samples, j is the index for the time domain, k is the index for the frequency domain, and j is the imaginary unit.
This linear sweep signal can be described by the following mathematical expression $f(t)=f_0+\frac{(f_1-f_0)}{t_1}t$. Where, t denotes time, f0 is the start frequency at time 0 and f1 the instantaneous frequency at time t1. Concretely, in our implementation we used f0 = 0Hz, f1 = 20kHz, t1 = 10s
Reviewer#1, Concern #4: 4) In the experimental part, due to the aging of equipment, the change of acoustic features leads to a significant decline in recognition rate. Although the dynamic update strategy is proposed, the redundancy problem of long-term dependence on other authentication methods is not solved.
Author response: We thank the reviewer for raising this issue. Indeed, this is a serious concern for most of the existing methods and in fact, most of the security standards from the industry already recommend redoing the authentication periodically in case of prolonged periods of inactivity.
The following text was added to the paper:
or to reauthenticate the device. In fact, this recommendation is already included in some security standards or draft recommendations from the industry, such as NIST SP 800-63B [63 ], OWASP [ 64 ], ISO/IEC 27001 [ 65 ] and ISO/IEC 27002 [66 ], etc.
Reviewer#1, Concern #5: 5) As shown in Figure 12, acoustic signal is distorted and identification fails completely at low temperature, but the author does not propose temperature compensation or multi-model switching scheme. In addition, as shown in Figure 10, the recognition rate drops to 0% when more than 160cm is exceeded, which causes the scheme to be less feasible in practice.
Author response: We thank the reviewer for the remark. The paper states that a multimodal authentication can be triggered if temperature is lower than 2° Celsius. Further investigation may require more complex, long-term experiments which we may consider as future work. Regarding the 160cm distance, we believe that this should be enough for an in-vehicle scenario since the space is limited by physical factors. Moreover, extending the authentication further can weaken the security level as the attack surface extends.
Reviewer#1, Concern #6: 6) Lack of further proof of security, how will it cope when malicious devices tamper with acoustic signatures through physical contact? For example, are adversarial attacks that currently only test simple recording playback resistant to generating adversarial examples or signal tampering?
Author response: We thank the reviewer for the remark. In case of device fingerprinting, the security of the methods is usually justified by empirical evidence. We were unable to find any formalism in the existing works that would provide clear mathematical bounds on effectiveness of this method. All existing works seem to be based on experimental results, similar to our work. Experiments regarding the adversary capability of impersonation are outlined in section 5.4. The possibility of an adversary overlapping with the current authentication is not discussed since this will lead to a DoS as the fingerprint will no longer match. More complex approaches, like Generative Adversarial Network are out of scope for the current communication but may be of interest for our future works.
Reviewer 2 Report
Comments and Suggestions for Authors
The authors address a very interesting and timely topic, and considering the rest of the article, the title is appropriately chosen and well-formulated.
In the introduction, the authors clearly demonstrate their reasons for pursuing this topic. The cited sources are already well-chosen and relevant to the article's subject. In the introduction, the authors mention the added value of the article—what they hoped to achieve with their research—which certainly facilitates further consideration of the authors' arguments.
The next chapter, "RElated Works," demonstrates the topic's timeliness. The authors demonstrate this with relevant cited articles on the topic.
The authors then present implementations of their idea. Figure 2 clearly demonstrates the concept, its understanding, and the verification of its validity. The description is comprehensive and thorough, providing a good foundation for understanding the theoretical foundations and the research environment.
The test studies presented later in the article appear very credible and interesting. The presented verification methods and corresponding graphs provide a good basis for the presented experimental data.
The description of the results is clear and comprehensive. Already in this section of the article, the authors include some discussion of the obtained results, which is certainly a plus for the presented argument. Although it is not necessary, I would suggest a separate chapter with a discussion of the presented results and solutions. Furthermore, I find the presented article interesting and worthy of publication.
Author Response
Reviewer#2, Concern #1:The authors address a very interesting and timely topic, and considering the rest of the article, the title is appropriately chosen and well-formulated.
In the introduction, the authors clearly demonstrate their reasons for pursuing this topic. The cited sources are already well-chosen and relevant to the article's subject. In the introduction, the authors mention the added value of the article—what they hoped to achieve with their research—which certainly facilitates further consideration of the authors' arguments.
The next chapter, "RElated Works," demonstrates the topic's timeliness. The authors demonstrate this with relevant cited articles on the topic.
The authors then present implementations of their idea. Figure 2 clearly demonstrates the concept, its understanding, and the verification of its validity. The description is comprehensive and thorough, providing a good foundation for understanding the theoretical foundations and the research environment.
The test studies presented later in the article appear very credible and interesting. The presented verification methods and corresponding graphs provide a good basis for the presented experimental data.
The description of the results is clear and comprehensive. Already in this section of the article, the authors include some discussion of the obtained results, which is certainly a plus for the presented argument. Although it is not necessary, I would suggest a separate chapter with a discussion of the presented results and solutions. Furthermore, I find the presented article interesting and worthy of publication.
Author response: We are grateful to the reviewer for the positive appreciation of our work and for the constructive remark regarding the paper structure. Based on the reviewer’s remark, in this revision we split the Experimental Results section in 2 separate sections. Section 5, Experimental Results, includes 2 subsections: 5.1. Off-line measurements and results and 5.2. Live measurements and results. Section 6. Limitations and Discussions, includes 3 subsections: 6.1. Threat model: device impersonation by replaying sounds, 6.2. Distance influence on the recognition rate and 6.3. Environment influence on the recognition rate.
Reviewer 3 Report
Comments and Suggestions for Authors
In this paper, authors present an Android-based system, AudioUnlock, that authenticates devices using unique acoustic signatures. The authors implement two one-class classifiers (OC-SVM and LOF), create a dataset of over 1,500 measurements across 23 devices, and test performance in both controlled and real-world automotive environments. They evaluate recognition rates under noise, impersonation attacks, distance variation, aging, and environmental conditions (temperature, altitude). Results show promising recognition rates (up to 90%+) and resilience to replay attacks, though accuracy decreases with distance and aging. The contribution lies in shifting from multi-class to one-class classifiers, making the system scalable to previously unseen devices.
The use of one-class classification for acoustic D2D authentication is interesting and practically motivated. However, presentation should be improved. Writing is generally clear, but too lengthy in places.
Some comments:
- Give url references inside text as reference.
- In abstract, add dataset size, recognition accuracy, and main limitation.
- Section 3: Implementation details are overly verbose (long description of microphones).
- Proofreading is needed, e.g., “bur rather” (p. 17) should be “but rather.” also “spikiness, etc
- Abstract line 1–4: simplify phrasing; avoid “physical imperfections in electronics…” and instead say “manufacturing-induced variations in microphones and speakers create unique acoustic fingerprints.”
- Abstract line 11–13: the automotive use-case appears suddenly; it should be introduced with motivation.
- In the introduction, move the MFA discussion later to avoid breaking the narrative.
- In the conclusion, highlight the most promising result (e.g., >90% recognition accuracy under certain conditions) and contrast with limitations.
- Figures 3–13 are informative but could be condensed; consider focusing on the most essential results.
Author Response
Reviewer#3, Concern #1: In this paper, authors present an Android-based system, AudioUnlock, that authenticates devices using unique acoustic signatures. The authors implement two one-class classifiers (OC-SVM and LOF), create a dataset of over 1,500 measurements across 23 devices, and test performance in both controlled and real-world automotive environments. They evaluate recognition rates under noise, impersonation attacks, distance variation, aging, and environmental conditions (temperature, altitude). Results show promising recognition rates (up to 90%+) and resilience to replay attacks, though accuracy decreases with distance and aging. The contribution lies in shifting from multi-class to one-class classifiers, making the system scalable to previously unseen devices.
The use of one-class classification for acoustic D2D authentication is interesting and practically motivated. However, presentation should be improved. Writing is generally clear, but too lengthy in places.
Author response: We thank the reviewer for the positive appreciation of our work.
Reviewer#3, Concern #2:- Give url references inside text as reference.
Author response: We thank the reviewer for the remark. We cited now all the URLs according to the mdpi template.
Reviewer#3, Concern #3: - In abstract, add dataset size, recognition accuracy, and main limitation.
Author response: We thank the reviewer for the remark. We added the following text in the abstract:
We created a dataset consisting of more than 5,000 measurements and achieved a recognition rate ranging from 50% to 100% for different devices under various conditions. These conditions also represent our limitations, such as distance, altitude, and component aging; however, we have proposed different solutions for the limitations that are part of our threat model.
Reviewer#3, Concern #4: - Section 3: Implementation details are overly verbose (long description of microphones).
Author response: We thank the reviewer for the remark. We removed some of the electrical details that we presented.
Reviewer#3, Concern #5: - Proofreading is needed, e.g., “bur rather” (p. 17) should be “but rather.” also “spikiness, etc
Author response: We thank the reviewer for the remark. We corrected these errors and further found others too, that we corrected as follows:
Line 256: ‘’spikiness” -> “spikeness”
Line 446: ‘bur rather” -> “but rather”’
Line 226: “theirs manufacturer” -> “their manufacturer”
Line 134: “Naive Baies” -> “Naive Bayes”
Line 115: “This works” -> “This work”
Lin 418: “Treath model” -> “Threat model”
Line 26: “manages” -> “manage”
Line 180: “electronical details” -> “electrical details”
Line 121: ”atmospherics pressure” -> “atmospheric pressure”
etc.
Reviewer#3, Concern #6: - Abstract line 1–4: simplify phrasing; avoid “physical imperfections in electronics…” and instead say “manufacturing-induced variations in microphones and speakers create unique acoustic fingerprints.”
Author response: We thank the reviewer for this excellent suggestion. We changed the sentence from the abstract as suggested:
due to manufacturing-induced variations in microphones and speakers.
Reviewer#3, Concern #7: - Abstract line 11–13: the automotive use-case appears suddenly; it should be introduced with motivation.
Author response: We thank the reviewer for the remark. In the Introduction section we further explained the automotive context, together with the motivation. The following text has been added to the introduction:
Modern-day cars still use classical keys for authentication, while only a few high-end manufacturers are releasing solutions that use smartphones. But existing initiatives such as the Car Connectivity Consortium [3], in which car and phone producers are developing industry standards and solutions to use smartphones as car keys, are promising for the future of smartphones within the car ecosystem. So far, the main solution released by the aforementioned collaboration is based on NFC and UWB but other options might be taken into account, a reason for which, more research on alternative sources for authentication may help.
Reviewer#3, Concern #8: - In the introduction, move the MFA discussion later to avoid breaking the narrative.
Author response: We thank the reviewer for this helpful remark. We have moved the MFA discussion to line 47, right after the presentation of the automotive context. In this way, the MFA section naturally follows the motivation, showing that, besides the automotive scenario, acoustic D2D authentication can also enhance MFA, without interrupting the narrative flow.
Reviewer#3, Concern #9: - In the conclusion, highlight the most promising result (e.g., >90% recognition accuracy under certain conditions) and contrast with limitations.
Author response: We thank the reviewer for the remark. We stated the most promising results. The following text has been added to conclusion:
with an average of 73.81% for the LOF algorithm and 75.48% for SVM, when running on smartphones
Reviewer#3, Concern #10: - Figures 3–13 are informative but could be condensed; consider focusing on the most essential results.
Author response: We thank the reviewer for the remark. Indeed, we combined three figures (previous figures 11, 12, 13) into one, and the new result is now presented in figure 11.
Round 2
Reviewer 1 Report
Comments and Suggestions for Authors
Thank you very much for considering my comments, the quality of the article has been improved.